# Peer mentorship to improve self-management of hip and knee osteoarthritis: a randomised feasibility trial

Anna M Anderson ,[1,2] Elizabeth C Lavender,[1] Esther Dusabe-Richards,[1] Teumzghi F Mebrahtu ,[1] Linda McGowan,[1] Philip G Conaghan ,[2] Sarah R Kingsbury,[2] Gerry Richardson,[3,4] Deborah Antcliff,[1,5] Gretl A McHugh[1]

[1]School of Healthcare, University of Leeds, Leeds, UK
[2]Leeds Institute of Rheumatic and Musculoskeletal Medicine and NIHR Leeds Biomedical Research Centre, University of Leeds, Leeds, UK
[3]Centre for Health Economics, University of York, York, UK
[4]NIHR Research Design Service for Yorkshire and the Humber, York, UK
[5]Physiotherapy Department, Bury Care Organisation, Northern Care Alliance NHS Group, Bury, UK

**Correspondence to**
Anna M Anderson;
A.Anderson@leeds.ac.uk

## ABSTRACT

**Objective**  To determine the feasibility of conducting a randomised controlled trial (RCT) of a peer mentorship intervention to improve self-management of osteoarthritis (OA).

**Design**  Six-month parallel group non-blinded randomised feasibility trial.

**Setting**  One secondary care and one primary care UK National Health Service Trust.

**Participants**  Fifty adults aged ≥55 years old with hip and/or knee OA.

**Interventions**  Participants were allocated 1:1 to the intervention or control group using an online randomisation service. Intervention group participants received usual care (information resources) and up to eight community-based self-management support sessions delivered by a peer mentor (trained volunteer with hip and/or knee OA). Control group participants received usual care only.

**Outcome measures**  Key feasibility outcomes were participant and peer mentor recruitment and attrition, intervention completion and the sample size required for a definitive RCT. Based on these feasibility outcomes, four success criteria for proceeding to a definitive RCT were prespecified. Patient-reported outcomes were collected via questionnaires at baseline, 8 weeks and 6 months.

**Results**  Ninety-six individuals were screened, 65 were eligible and 50 were randomised (25 per group). Of the 24 participants who commenced the intervention, 20 completed it. Four participants did not complete the 6-month questionnaire. Twenty-one individuals were eligible for the peer mentor role, 15 were trained and 5 withdrew prior to being matched with a participant. No intervention-related harms occurred. Allowing for 20% attrition, the sample size required for a definitive RCT was calculated as 170 participants. The intervention group showed improvements in self-management compared with the control group.

**Conclusions**  The feasibility outcomes achieved the prespecified criteria for proceeding to an RCT. The exploratory analyses suggest peer mentorship may improve OA self-management. An RCT of the OA peer mentorship intervention is therefore warranted with minor modifications to the intervention and trial procedures.

**Trial registration number**  ISRCTN:50675542.

## Strengths and limitations of this study

► This randomised feasibility trial is the first study to develop a novel peer mentorship intervention specifically focused on improving self-management of hip and knee osteoarthritis (OA).

► Extensive patient and public involvement ensured the intervention and trial procedures were tailored to the needs of individuals with OA.

► A comprehensive range of feasibility outcomes were assessed, providing valuable information for designing a future definitive randomised controlled trial.

► Validated patient-reported outcome measures were administered; however, the results must be interpreted cautiously because the trial was not powered to detect statistically significant differences.

► A key limitation was that the majority of participants were recruited through a physiotherapy service and hence had already received some self-management support.

## INTRODUCTION

Osteoarthritis (OA) is one of the most prevalent musculoskeletal conditions, with hip and knee OA affecting over 300 million individuals worldwide.[1] Furthermore, the prevalence of OA is increasing due to the ageing population and rising obesity levels.[2] Individuals with OA often experience severe pain, impaired function and reduced quality of life.[3] OA can have a profound psychosocial impact[4] and results in substantial economic burden.[5]

National and international guidelines emphasise that patient education, self-management strategies and exercise are core elements for managing OA.[6] However, implementation of OA guidelines is currently poor.[7] Patients report receiving insufficient information about OA management[8] and having a limited understanding of the condition.[4 9] This negatively impacts patients'

health behaviours.[4 9] A need for tailored interventions to support OA self-management has therefore been highlighted.[10]

Peer support interventions are an established approach for supporting chronic condition self-management.[11 12] Various peer support models have been described, such as peer-led group programmes and peer mentorship.[13] The latter involves a trained individual with a particular health condition (the 'peer mentor') providing one-to-one support to another individual with the same condition.[13] This approach is likely to be particularly valuable for individuals with OA due to the heterogeneous nature of the condition, which means tailored support is crucial.[10 14]

Previous studies have highlighted the value of OA interventions incorporating peer support.[15] However, no previous studies have explored an OA peer mentorship intervention. Although such an intervention offers multiple potential benefits, peer mentorship interventions can present feasibility/acceptability issues, such as high peer mentor attrition.[16] Therefore, this feasibility trial aimed to develop and trial a peer mentorship intervention to improve OA self-management. Its key objectives were to determine the feasibility of conducting a randomised controlled trial (RCT) of the OA peer mentorship intervention in terms of: the feasibility/ acceptability of the intervention and trial procedures; participant recruitment and retention; questionnaire completion rates; generating the sample size required for a definitive RCT; and estimating the intervention costs. Additionally, the potential impact of the intervention on patient-reported outcomes and resource use was explored.

## METHODS

This was a 6-month parallel group randomised feasibility trial. A nested qualitative study was included (reported elsewhere). The trial was prospectively registered and conducted between 1 September 2017 and 16 February 2020. The trial is reported according to the CONSORT 2010 extension for randomised pilot and feasibility trials[17] (online supplemental tables 1 and 2). The development and feasibility testing of the OA peer mentorship intervention was guided by the Medical Research Council guidance on developing and evaluating complex interventions.[18]

### Trial procedures and participants
#### Recruitment and consent
Potential participants were identified from rheumatology and orthopaedic clinics of one secondary care National Health Service (NHS) Trust and physiotherapy clinics and electronic records of one primary care NHS Trust. Both are large Trusts in Northern England. The initial approach was made via clinical staff during clinic appointments or through an invitation letter. Individuals interested in participating were provided with further details about the trial and screened for eligibility by a researcher at their clinic appointment or via telephone.

All participants provided written informed consent at their clinic appointment or during a baseline visit from a researcher. The latter took place in the participant's home or another private location of the participant's choice.

### Eligibility criteria
Inclusion criteria were: (1) aged ≥55 years old and (2) clinician-confirmed diagnosis of hip and/or knee OA.[19] Exclusion criteria were: (1) presence of inflammatory arthritis (including gout and rheumatoid arthritis), (2) serious health conditions that would prevent participation and (3) listed for hip/knee replacement.

### Sample size
Recommendations for pilot studies suggest 20 participants per arm is acceptable assuming at worst a small effect size (Cohen's d=0.2) for a continuous outcome and 80% power.[20] Allowing for 20% attrition,[21] the sample size was set at 25 participants per arm.

### Randomisation
Participants were randomised to the intervention or control group with 1:1 allocation using the Sealed Envelope online randomisation service.[22] This was set up by an independent statistician to generate blocked randomisation with varying block lengths, stratified according to educational level. The researcher performing the randomisation was unable to access the allocation codes. Given the nature of the intervention, participant blinding was not possible.

### Intervention group
Intervention group participants received usual care and the OA peer mentorship intervention. Usual care consisted of information resources (an OA booklet from Arthritis Research UK (now Versus Arthritis)[23] and a handout about local services/support groups/activities). The intervention group participants received the information resources during their initial mentorship session and were offered the opportunity to discuss the resources with their peer mentor.

### Control group
Control group participants received usual care only. The control group participants received the information resources during their baseline researcher visit and were offered the opportunity to discuss the resources with the researcher.

### OA peer mentorship intervention
#### Intervention development
The OA peer mentorship intervention was developed in two stages. The first stage included a rapid review of published primary studies investigating one-to-one peer support interventions. The review aimed to identify: the range of methods and approaches used in delivering peer

support interventions; training and support approaches for peer mentors involved in intervention delivery; and challenges of developing and implementing face-to-face peer support to improve self-management of long-term conditions. Using Medline, CINAHL and PsycInfo, a search of the literature between 2007 and 2018 was undertaken using the terms and synonyms 'peer support', 'long term condition' and 'intervention'. Thirteen papers were included.[16 24–35] The findings highlighted the importance of encouraging a person-centred approach and retaining flexibility within the peer mentorship sessions. There was little information about 'matching' the mentee with the peer mentor, although some studies based this on factors such as gender and age. There were issues around recruitment of peer mentors and a need to provide support and guidance to peer mentors throughout the intervention.

A preliminary version of the OA peer mentorship intervention was developed based on the rapid review findings and the following sources:

► Guidelines on self-management and OA from organisations such as the UK National Institute for Health and Care Excellence and the UK charity Versus Arthritis.
► The 'Staying Connected Programme': an arthritis self-management programme previously run by Arthritis Care Northern Ireland (now Versus Arthritis Northern Ireland).[36]
► Input from project team members, including a consultant rheumatologist, a health psychologist and a physiotherapist specialising in activity pacing/chronic pain management.

The above-mentioned sources were also used to develop a draft peer mentor educational resource pack. The pack was designed to supplement peer mentor training sessions and be used a resource during mentorship sessions. The pack included a range of handouts that peer mentors could give to participants.

The second stage of the development process consisted of expert review sessions conducted with the following key stakeholders: study patient and public involvement (PPI) members (n=2); other older individuals with OA (n=5); health professionals (n=4); voluntary/community organisation representatives (n=4); and researchers (n=2). Review sessions were conducted face to face (group, paired or individual meetings), via telephone or via email. The stakeholders were provided with information about the provisional OA peer mentorship intervention and a copy of the draft educational pack. Stakeholders' opinions of these were assessed using a pro forma.

Key refinements made based on the expert review sessions included:

► Peer mentors were encouraged not to cover too many topics in the first mentorship session to allow more time for developing rapport and managing participant expectations.
► Participants were provided with notebooks for recording goals, reflections and additional notes.

► The language used in the educational pack was simplified.
► Additional infographic handouts and further information on falls, local authority services and emotional well-being were added to the educational resource pack.

## Finalised intervention

The finalised OA peer mentorship intervention aimed to improve participants' health outcomes through increasing their engagement with self-management behaviours. Figure 1 presents a logic model of the intervention, including the proposed mechanisms of action. The intervention involved up to eight 1-hour self-management support sessions delivered by a trained peer mentor. During the sessions, the peer mentor provided guided support that incorporated multiple behaviour change techniques (BCTs) and covered a combination of core and optional topics (figure 1). In line with a person-centred approach, the implementation of BCTs and the choice and order of topics covered was flexible and participant led. However, peer mentors were encouraged to cover all core topics at least once and set/review goals with the participant weekly. Online supplemental table 3 provides examples of the implementation of the key BCTs employed.

The peer mentorship sessions took place approximately weekly in the participant's home or another private location of their choice. A person-centred approach was emphasised throughout.[37 38] Therefore, the number of sessions and scheduling were flexible.

The volunteer coordinator attended the start of each initial session to introduce the participant and peer mentor, remind them of the trial and intervention aims and answer questions. The remainder of the initial session and all subsequent sessions were undertaken by the mentor alone. The volunteer coordinator contacted the participant and peer mentor following the initial session to check they were happy to continue with subsequent sessions with their mentorship match. Peer mentors completed a 'session summary' in writing following each session, detailing the topics covered during that session, any challenges encountered and their reflections on the progress made.

## Peer mentor eligibility and recruitment

Peer mentors were trained volunteers aged ≥50 years old with hip and/or knee OA who were able to travel independently. Multiple approaches were used to recruit peer mentors, including printed/social media advertisements and sharing trial information at local support/activity groups.

Potential peer mentors were asked to complete an application form, supply two references and take part in a telephone interview with the volunteer coordinator. Those assessed as suitable were invited to attend a compulsory 2-day training event. Prior to being matched with a participant, all peer mentors were required to complete

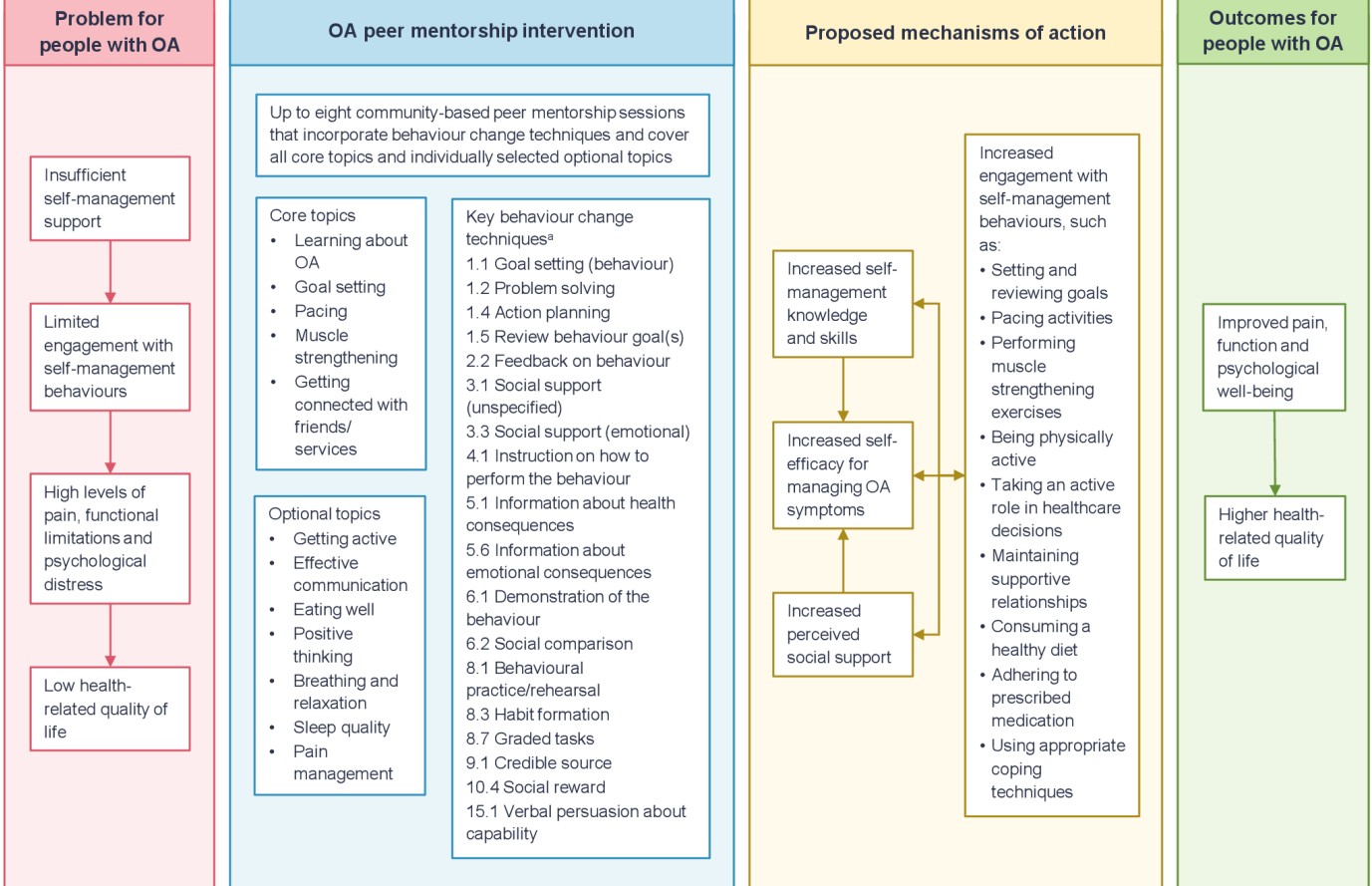

**Figure 1** OA peer mentorship intervention logic model. [a]Behaviour change techniques are coded using the Behaviour Change Technique Taxonomy version 1.[60] OA, osteoarthritis.

enhanced Disclosure and Barring Service (criminal record) checks[39] for safeguarding purposes.

### Peer mentor training and matching
Three training events were held due to varying peer mentor availability and staggering of the mentor recruitment, which meant it was not possible to train all mentors in one event. The first training event was provided by two external facilitators from Arthritis Care Northern Ireland and a trial team member (physiotherapist specialising in activity pacing/chronic pain management). The remaining events were provided by three trial team members (volunteer coordinator, the physiotherapist who facilitated the first training event and another musculoskeletal physiotherapist). Feedback from the first event was used to refine the content/delivery of the subsequent events.

The training events involved presentations and interactive activities covering OA self-management topics, mentorship skills and the practicalities of the peer mentor role (online supplemental table 4).

Participant/mentor matching was undertaken by the volunteer coordinator and the researcher who completed the baseline researcher visit. The initial intention was to base matching on location, gender preference, age and OA site(s). However, due to peer mentors' preferences

and discrepancies in the availability of intervention group participants and trained peer mentors, these criteria were relaxed.

### Data collection and analysis
#### Feasibility outcomes
This trial's primary focus was on assessing the feasibility outcomes specified in table 1. Four associated success criteria for proceeding to a definitive RCT were prespecified based on relevant guidance[40] (table 1).

#### Patient-reported outcomes
Patient-reported outcomes were assessed using paper self-report questionnaires administered at baseline, 8 weeks and 6 months. The baseline questionnaires were administered by a researcher or the volunteer coordinator. The majority of baseline questionnaires were administered after the participant had been informed of their group allocation. This approach was chosen due to the fluctuating availability of trained peer mentors. This meant there was sometimes a delay between participants being allocated to the intervention group and being able to commence their mentorship sessions. Therefore, most intervention group participants completed the baseline questionnaire with the volunteer coordinator immediately prior to their initial mentorship session to avoid a

**Table 1** Feasibility outcomes and success criteria for proceeding to a definitive randomised controlled trial

| Feasibility outcome | Details of feasibility outcome | Success criterion |
|---|---|---|
| Peer mentor recruitment rate | Number of peer mentors trained divided by the number individuals who were eligible for the peer mentor role | 1. Demonstration that the intervention can be delivered in practice and is acceptable to participants and peer mentors* |
| Peer mentor attrition rate | Number of peer mentors who were not matched with a participant divided by the number of peer mentors trained | |
| Intervention completion rate | Number of participants who completed at least three peer mentorship sessions divided by the number of participants who commenced the intervention | |
| Participant recruitment rate | Number of participants randomised divided by the number of individuals who were eligible to participate | 2. ≥60% of eligible individuals recruited |
| Attrition rate overall and in each group | Number of participants who did not complete the 6-month questionnaire divided by the number of participants randomised | 3. <20% participant attrition with no evidence of attrition bias |
| Sample size required for a definitive RCT | Sample size calculation using the feasibility trial PIH scale scores assuming: 5% significance level; minimum power of 0.8; and 1:1 allocation ratio. The PIH scale scores were used because the PIH scale was chosen as the provisional primary outcome measure. | 4. Calculation of a sample size that is achievable in a main trial |
| Questionnaire completion rate | Number of questionnaires completed divided by the number of participants provided with the questionnaires | None specified for these feasibility outcomes |
| Intervention fidelity | Content analysis of the mentors' session summaries | |
| Cost of delivering the mentor training event | Estimation based on the events delivered by the trial team members accounting for: staff salary, national insurance and pension contributions; resources; refreshments; and peer mentor travel expenses; but not venue hire, estates and indirect costs | |
| Cost of delivering the intervention | Estimation based on the volunteer coordinator and peer mentor costs, excluding the training costs | |

*No minimum/maximum rates were prespecified as acceptable for this success criterion.
PIH scale, revised 12-item Partners In Health scale; RCT, randomised controlled trial.

delay between completion of the baseline questionnaire and commencement of their mentorship sessions.

The 8-week and 6-month questionnaires were administered and returned via post. Non-responders were followed up with a second questionnaire posting 2 weeks later and a telephone call 4 weeks later.

The questionnaires covered participants' sociodemographic and clinical characteristics and included validated tools assessing the biomedical/psychosocial outcomes specified below.

### Provisional primary outcome measure
▶ Revised 12-item Partners in Health (PIH) Scale: 12-item scale that assesses chronic condition self-management.[41] Each item is scored on a 9-point Likert scale (0–8). Scores are summed to give a total score (0–96). Higher scores indicate higher self-management knowledge/behaviours.

### Provisional secondary outcome measures
▶ Multidimensional Scale of Perceived Social Support (MSPSS): 12-item scale that assesses perceived social support from a significant other, family and friends.[42] Each item is scored on a 7-point Likert

scale (1–7), and the mean for all items is calculated. Higher scores indicate greater perceived social support.
▶ Western Ontario and McMaster's University Osteoarthritis Index Likert version (WOMAC 3.1): 24-item disease-specific scale with three subscales that assess pain (5 items), stiffness (2 items) and physical function (17 items).[43] Each item is scored on a 5-point Likert scale (0–4). Scores are summed to give a total score (0–96) and subscale scores (pain: 1–20; stiffness: 0–8; physical function: 0–68). Higher scores indicate more severe problems.
▶ Eight-item Arthritis Self-Efficacy Scale English version: 8-item scale that assesses self-efficacy for managing arthritis symptoms.[44–46] Each item is scored on a 10-point Likert scale (1–10), and the mean for all items is calculated. Higher scores indicate greater self-efficacy.
▶ Hospital Anxiety and Depression Scale (HADS): 14-item scale with two subscales that assess symptoms of anxiety (7 items) and depression (7 items).[47] Each item is scored on a 4-point Likert scale (0–3). Scores are summed to give a total score (0–42) and subscale

scores (0–21). Higher scores indicate more severe symptoms.

▶ EQ-5D-5L: Descriptive system and visual analogue scale (VAS) that assess general health status.[48 49] The descriptive system includes five dimensions, each of which has five response levels. Each response is converted to a single-digit number. The numbers for each of the five dimensions can then be converted to a single index value anchored at 0 (a state equivalent to dead) and 1 (full health). The VAS consists of a single score on a scale from 0 (worst health imaginable) to 100 (best health imaginable).

The 8-week and 6-month questionnaires included sections on healthcare and community resource use, adapted from the Client Services Receipt Inventory.[50]

Missing data were addressed in line with guidance for the relevant outcome measurement tool. The quantitative questionnaire responses were analysed using STATA (V.15). Given the trial was not powered to detect statistically significant differences, the analyses were predominantly descriptive. However, exploratory analyses of the outcome measure data and healthcare resource use data were performed. The outcome scores were tested for normality at the 5% significance level.[51] Between-group comparisons were made using analysis of covariance models and, where the scores were significantly skewed, quantile regression models were used.[52]

## Patient and public involvement

PPI played a key role in this trial as follows:

▶ During the trial development stage, six individuals with OA were consulted for research ideas, and eight members of a local musculoskeletal PPI group participated in a group discussion. These patient representatives emphasised that one-to-one mentorship would be preferable to a group-based approach and felt the questionnaire burden was appropriate.

▶ Three patient representatives became 'study PPI members' and assisted with the conduct of the study, reviewing lay materials and finalising the dissemination plans. These PPI members were also invited to attend peer mentorship training sessions.

▶ Two study PPI members and five other individuals with OA helped refine the provisional intervention and educational resources through expert review sessions.

▶ The peer mentors were all trained volunteers with hip and/or knee OA. Peer mentors were encouraged to give regular informal feedback about their involvement in the trial, as well as being invited to participate in a nested qualitative study (reported elsewhere).

# RESULTS
## Feasibility outcomes

### Participant flow and intervention completion
Participant recruitment took place between 22 November 2018 and 30 May 2019. The final questionnaire was completed on the 16 February 2020. At least 345 individuals were approached (figure 2). Ninety-six individuals were screened, of whom 65 were eligible. The most common reason for ineligibility was being listed for joint replacement (n=8). Fifty individuals were randomised. Therefore, the participant recruitment rate was 77%. The most common reason for declining participation was insufficient time/other commitments (n=7). Most participants were recruited through the primary care physiotherapy department (n=44).

One intervention group participant withdrew due to ill-health prior to completing the baseline questionnaire and commencing the intervention. Of the 24 participants who commenced the intervention, 20 completed at least three mentorship sessions (figure 2). The intervention completion rate was therefore 83%. Reasons for discontinuing the intervention largely related to participants feeling the intervention was not relevant to their needs, for example, due to having mild OA symptoms and/or feeling they did not require additional self-management support. The mean number of mentorship sessions received was 5.79 (SD=2.25; median=7). No harms/unintended effects related to the intervention occurred.

One intervention participant who discontinued the intervention after two mentorship sessions withdrew (no reason provided). One additional intervention participant who discontinued the intervention after two mentorship sessions did not complete the follow-up questionnaires. One control group participant did not complete the follow-up questionnaires. A family member of this control group participant returned his 6-month questionnaire indicating he had died. The attrition rates were therefore: 8% overall, 12% in the intervention group and 4% in the control group.

### Questionnaire completion
Among the participants who were provided with the questionnaires, the questionnaire completion rate for both groups was 100% at baseline and 96% at 8 weeks and 6 months (figure 2). Minor issues were noted with the resource use sections. Some participants duplicated details, for example, by recording the same physiotherapy appointments under 'Hospital services' and 'Services outside the hospital'. Additionally, the 'day activities' reported by participants largely related to their hobbies, such as yoga and dancing, rather than social/community support activities.

### Peer mentor flow and characteristics
Peer mentor recruitment took place between May 2018 and January 2019. Thirty-two individuals enquired about the role, of whom 21 were eligible (online supplemental figure 1). Five eligible individuals were unable to attend the training events, one withdrew due to ill-health and 15 were trained. The peer mentor recruitment rate was therefore 71%. Five trained mentors withdrew prior to being matched, most commonly due to ill-health (n=3). The peer mentor attrition rate was therefore 33%. The

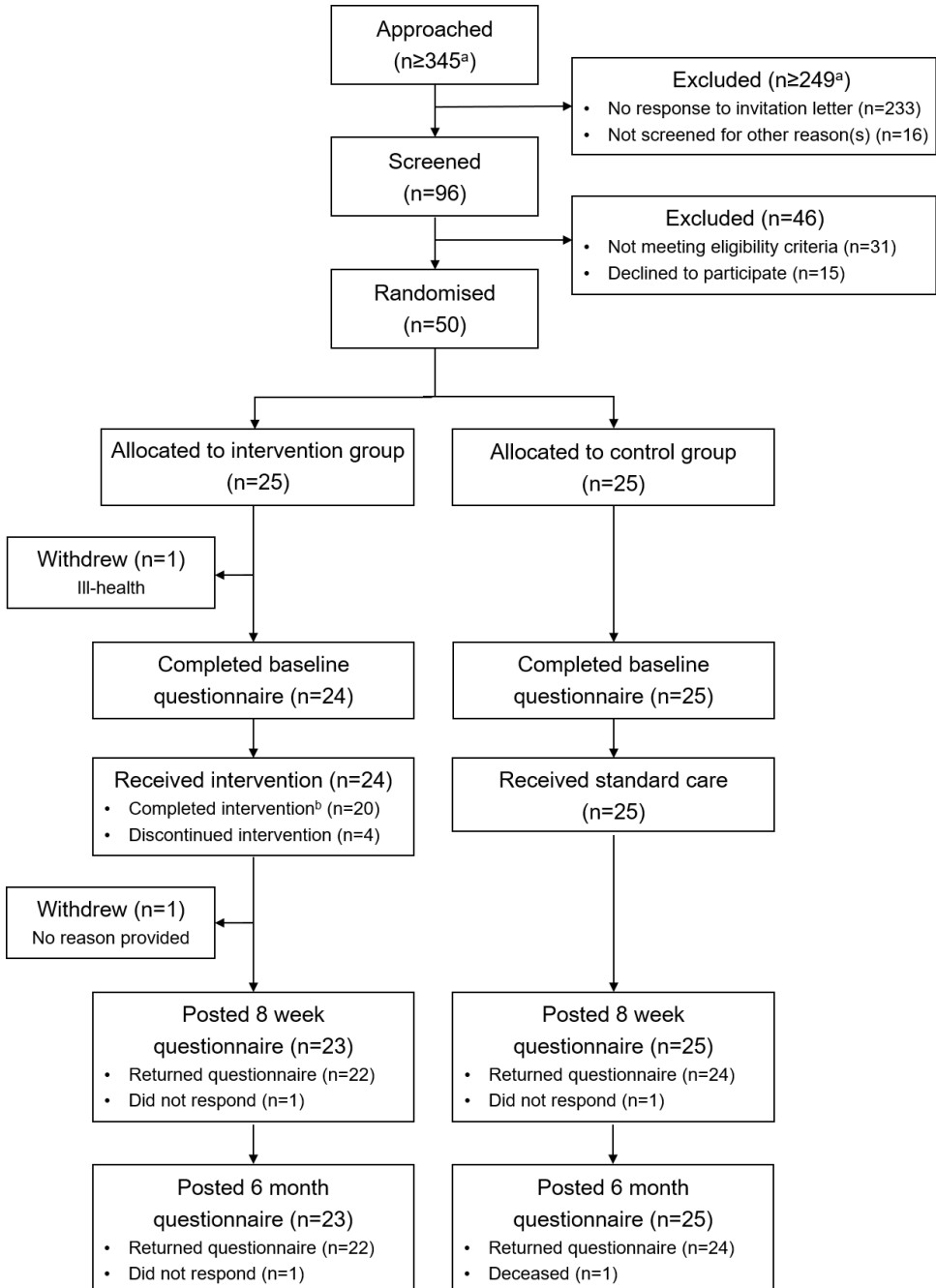

**Figure 2** Participant flow diagram. [a]Minimum number of individuals approached/excluded. Additional individuals may have been approached/excluded because the number of individuals approached through physiotherapy clinics was not recorded to minimise the administrative burden on physiotherapists. [b]Intervention completion was defined as completion of at least three peer mentorship sessions.

most successful peer mentor recruitment approach was a local community magazine advertisement, which led to recruitment of five active peer mentors.

Nine active peer mentors were female and one was male. Their mean age was 68 years old (SD=5.5; range=57–75). All had been living with hip and/or knee OA for at least 3 years. Nine were retired, although three had only recently stopped work, and one worked full-time. Each active peer mentor supported between one and four participants (online supplemental table 1).

**Session summary data**

The session summaries indicated the following topics were covered by almost all participant/mentor dyads and were the most frequently covered overall: learning about OA; goal setting; pacing; muscle strengthening; and pain management. In addition, getting active and eating well were frequently covered.

The least frequently covered topics were: getting connected with friends/services; and effective communication. Correspondingly, these topics were covered by

**Table 2** Participant baseline characteristics

| Characteristics | Intervention n=24 | Control n=25 |
|---|---|---|
| Age in years, mean (SD) | 70.0 (8.6) | 69.3 (8.1) |
| Men, n (%) | 10 (41.7) | 4 (16.0) |
| Ethnicity, white, n (%) | 21 (87.5) | 24 (96.0) |
| Further education*, n (%) | 15 (62.5) | 15 (60.0) |
| Employed, n (%) | 7 (29.2) | 5 (20.0) |
| Retired, n (%) | 17 (70.8) | 20 (80.0) |
| Body mass index in kg/m$^2$, mean (SD) | 27.5 (6.3) | 28.5 (5.5) |
| Duration of arthritis diagnosis in years, mean (SD) | 4.7 (5.2) | 5.3 (6.2) |
| At least one hip affected by arthritis, n (%) | 14 (58.3) | 13 (52.0) |
| At least one knee affected by arthritis, n (%) | 22 (91.7) | 21 (84.0) |
| VAS for current pain due hip/knee arthritis, mean (SD) | 52.8 (26.8) | 48.6 (26.3) |
| VAS for ability to cope with/manage arthritis in general, mean (SD) | 66.2 (21.8) | 64.4 (21.8) |
| Number of joints affected by pain for >6 weeks in the last 3 months, median (IRQ) | 5 (4) | 3 (3) |
| Number of comorbidities†, n (%) | | |
| 0 | 8 (33.3) | 6 (24.0) |
| 1 | 5 (20.8) | 10 (40.0) |
| 2 | 5 (20.8) | 3 (12.0) |
| ≥3 | 6 (25.0) | 6 (24.0) |

*Further education was defined as any formal education undertaken once the participant was aged over 16 years old.
†Comorbidities include: diabetes; asthma; bronchitis; gastrointestinal problems; angina/heart problems; high blood pressure; depression; and anxiety.
VAS, Visual Analogue Scale (0–100 mm).

only six and eight dyads, respectively. Positive thinking and sleep quality were also rarely covered.

### Intervention costs
The estimated cost of delivering the training event to five peer mentors (mean number of mentors per event) was £239 per mentor (online supplemental table 5). The estimated cost of delivering the intervention based on the mean number of 5.79 mentorship sessions was £274 per participant (online supplemental table 6).

### Sample size calculation
Based on the assumptions stated in table 1, between-groups variance and within-groups variance of the PIH scale at 6 months of 369.1 and 6237.1, respectively, and detecting a difference in the PIH scale of 4.4 (effect size at 6 months), the sample size required for a definitive RCT was calculated as 136 participants (68 per group). Retaining the same attrition rate as for the feasibility trial (20%) would give a required sample size of 170 (85 per group).

### Patient-reported outcomes
Participant baseline characteristics
Table 2 provides a summary of participants' baseline sociodemographic and clinical characteristics. Additional details are available in online supplemental table 7.

### Provisional primary outcome
The mean PIH scores at baseline were 73.9 and 76.5 for the intervention and control groups, respectively (table 3). After 8 weeks, the mean PIH score improved in the intervention group (mean change=2.6; SD=13.2) and deteriorated in the control group (mean change= −7; SD=12.3). Thus, the intervention group participants appeared to have better self-management knowledge/behaviours than the control group participants at 8 weeks (effect size=8.3; 95% CI 2.2 to 14.4) (table 4). At 6 months, the intervention group's mean PIH score was similar to the group's score at 8 weeks, while the control group's mean PIH score increased but remained lower than at baseline. Overall, the intervention group had higher PIH scores than the control group after 6 months (effect size=4.4; 95% CI −2.8 to 11.6) (table 4).

### Provisional secondary outcomes
There were no significant between-group differences for any of the secondary outcomes at 8 weeks or 6 months. However, the mean WOMAC and HADS scores were lower in the intervention group than the control group at baseline and both follow-up time-points.

### Resource use
Healthcare resource use between baseline and 6 months was largely similar between groups. The main differences

**Table 3** Outcome measurement scores at baseline, 8 weeks and 6 months

| Outcome measure | Baseline means (SD) | | 8-week means (SD) | | 6-month means (SD) | |
| --- | --- | --- | --- | --- | --- | --- |
| | Intervention (n=22) | Control (n=24) | Intervention (n=22) | Control (n=24) | Intervention (n=22) | Control (n=24) |
| PIH scale | 73.9 (16.3) | 76.5 (14.3) | 76.5 (14.5) | 69.5 (10.6) | 76.0 (13.4) | 73.0 (15.0) |
| MSPSS | 5.3 (1.1) | 5.7 (0.98) | 4.9 (1.6) | 5.2 (1.6) | 4.9 (1.3) | 5.4 (1.4) |
| WOMAC total* | 32.6 (18.8) | 41.9 (19.5) | 34.0 (17.8) | 43.5 (19.4) | 34.7 (18.5) | 41.3 (17.8) |
| WOMAC pain* | 6.8 (4.5) | 9 .0 (4.2) | 7.5 (3.9) | 9.6 (4.1) | 7.7 (4.0) | 8.9 (3.2) |
| WOMAC stiffness* | 3.7 (1.6) | 4.3 (1.9) | 3.3 (1.3) | 4.3 (1.6) | 3.4 (1.3) | 4.0 (1.7) |
| WOMAC function* | 22.1 (14.0) | 28.5 (14.6) | 23.2 (13.2) | 29.6 (15.3) | 23.6 (14.8) | 28.3 (13.5) |
| ASES-8 | 6.0 (1.9) | 5.9 (1.7) | 5.5 (2.2) | 6.5 (2.4) | 6.3 (1.6) | 5.3 (2.5) |
| HADS total* | 9.4 (4.4) | 11.3 (7.2) | 10.5 (5.7) | 12.7 (8.3) | 10.7 (6.2) | 12.8 (8.4) |
| HADS anxiety* | 5.3 (3.1) | 6.7 (4.3) | 6.3 (3.4) | 7.3 (4.8) | 5.9 (3.9) | 7.5 (4.9) |
| HADS depression* | 4.1 (2.0) | 4.6 (3.6) | 4.3 (3.1) | 5.4 (4.0) | 4.8 (2.7) | 5.3 (4.0) |
| EQ-5D-5L index‡, † | 0.723 (0.139) | 0.645 (0.170) | 0.691 (0.099) | 0.647 (0.249) | 0.687 (0.088) | 0.647 (0.183) |
| EQ-5D-5L VAS† | 70 (15.0) | 75 (20.0) | 70 (25.0) | 70 (22.5) | 75 (20.0) | 70 (23.5) |

*Higher scores indicate more severe problems.
†Median (IQR).
‡EQ-5D-5L index value control group n=23 due to missing data.
ASES-8, 8-item Arthritis Self-Efficacy Scale English version; HADS, Hospital Anxiety and Depression Scale; MSPSS, Multidimensional Scale of Perceived Social Support; PIH scale, revised 12-item Partners in Health Scale; VAS, Visual Analogue Scale; WOMAC, Likert version of the Western Ontario and McMaster's University Osteoarthritis Index.

were more overnight hospital stays in the control group (mean difference between groups=−0.91; 95% CI −1.78 to −0.04), and more GP practice nurse visits in the intervention group (mean difference between groups=0.93; 95% CI 0.15 to 1.71) (online supplemental table 8). No participants reported attending a day centre.

**Table 4** Outcome measurements score changes and effect of the intervention at 8 weeks and 6 months

| Outcome measure | Changes at 8 weeks | | Effect size at 8 weeks, mean (95% CI)* | Changes at 6 months | | Effect size at 6 months, mean (95% CI)* |
| --- | --- | --- | --- | --- | --- | --- |
| | Intervention | Control | | Intervention | Control | |
| PIH scale | 2.6 (13.2) | −7.0 (12.3) | 8.3 (2.2 to 14.4) | 2.1 (10.6) | −3.5 (16.6) | 4.4 (−2.8 to 11.6) |
| MSPSS | −0.28 (1.4)† | −0.25 (1.2)† | 0.0 (−0.80 to 0.80)‡ | −0.42 (0.93)† | −0.06 (0.6)† | −0.40 (−0.84 to 0.03)‡ |
| WOMAC total§ | 3.0 (19)† | 5.0 (22)† | −2.6 (−10.2 to 4.9)‡ | 2.1 (8.6) | −0.59 (19.1) | −0.65 (−8.9 to 7.6) |
| WOMAC pain§ | 0.64 (2.6) | 0.63 (3.7) | −0.75 (−2.5 to 1.0) | 0.91 (2.4) | −0.09 (3.8) | 0.06 (−1.6 to 1.7) |
| WOMAC stiffness§ | −0.36 (1.5) | −0.04 (1.1) | −0.59 (−1.3 to 0.08) | −0.27 (1.8) | −0.29 (1.2) | −0.31 (−1.1 to 0.46) |
| WOMAC function§ | 1.1 (7.8) | 1.0 (10.4) | −1.3 (−6.7 to 4.1) | 1.5 (7.8) | −0.21 (15.4) | −0.8 (−7.7 to 6.0) |
| ASES-8 | 0.48 (1.6) | −0.04 (2.25) | 0.64 (−0.51 to 1.8) | 0.69 (3.0)† | 0.0 (3.3)† | 0.34 (−1.3 to 1.9)‡ |
| HADS total§ | 1.12 (3.8) | 1.38 (4.5) | −0.27 (−2.8 to 2.3) | 1.33 (3.4) | 1.5 (4.3) | −0.01 (−2.4 to 2.4) |
| HADS anxiety§ | 0.95 (2.8) | 0.58 (2.1) | 0.26 (−1.2 to 1.7) | 0.59 (2.4) | 0.79 (2.6) | −0.23 (−1.8 to 1.3) |
| HADS depression§ | 0.20 (2.4) | 0.79 (3.4) | −0.74 (−2.5 to 1.0) | 0.74 (1.6) | 0.67 (2.5) | 0.03 (−1.3 to 1.3) |
| EQ-5D-5L index¶ | −0.057 (0.20) | 0 (0.12) | −0.06 (−0.18 to 0.06) | −0.053 (0.14) | 0 (0.19) | −0.04 (−0.12 to 0.04) |
| EQ-5D-5L VAS | −2.5 (20)† | 0 (10)† | 0 (−10.8 to 10.8)‡ | 0 (20)† | −5 (21)† | 6.7 (−4.6 to 18.0)‡ |

*Analysis of covariance models adjusted for baseline outcomes.
†Median (IQR).
‡Median (95% CIs).
§Higher scores indicate more severe problems.
¶EQ-5D-5L index value control group n=23 due to missing data.
ASES-8, 8-item Arthritis Self-Efficacy Scale English version; HADS, Hospital Anxiety and Depression Scale; MSPSS, Multidimensional Scale of Perceived Social Support; PIH scale, revised 12-item Partners in Health Scale; WOMAC, Likert version of the Western Ontario and McMaster's University Osteoarthritis Index.

Two intervention group participants and four control group participants reported receiving help with daily activities.

## DISCUSSION

This feasibility trial aimed to develop and trial a peer mentorship intervention to improve OA self-management. All four prespecified success criteria for proceeding to a definitive RCT (table 1) were achieved. The peer mentor recruitment rate (71%), peer mentor attrition rate (33%) and intervention completion rate (83%) all compare favourably with previous peer mentorship studies.[16 32 53 54] This suggests the OA peer mentorship intervention can be delivered in practice and is acceptable to participants and peer mentors. However, some participants discontinued the intervention due to having mild OA symptoms and feeling they did not need self-management support. This could be overcome by specifying a minimum symptom severity threshold and/or a maximum arthritis-related self-efficacy threshold in the eligibility criteria to prevent inclusion of individuals who have mild symptoms and/or already feel confident about managing their symptoms.

The participant recruitment rate was 77% and hence well above the prespecified minimum threshold of 60%. However, this recruitment rate did not account for individuals lost prior to screening. The overall participant attrition rate at 6 months was 8% and hence well below the prespecified maximum threshold of 20%. This suggests that additional strategies to improve participant recruitment and retention would not be needed for a future RCT. Allowing for 20% attrition, the sample size required for a definitive RCT was calculated as 170 participants (85 per group). Assuming each peer mentor supports two participants, this would require 43 mentors. This target RCT sample size is achievable. However, only 10 active peer mentors were recruited during this trial. Additional strategies would therefore be needed to recruit sufficient mentors. These could include using multiple recruitment sites across a wider geographical area, recruiting mentors over a longer time-period, using snowballing recruitment with previously trained peer mentors and optimising recruitment through media outlets, such as magazine advertisements.

The findings for the remaining feasibility outcomes were also encouraging. The questionnaire completion rates were consistently high (96%–100%). However, minor modifications to the resource use sections would be beneficial to avoid participants duplicating details/ reporting unnecessary details. The session summary data indicated intervention fidelity was good overall, with most participant/mentor dyads covering four of the five core topics. The remaining core topic, getting connected with friends/services, was only covered by six dyads. This may be because the participants already had good social support networks, as suggested by their relatively high MSPSS scores. Additionally, most dyads covered the optional topic of pain management. Therefore, changing

getting connected to an optional topic and pain management to a core topic warrants consideration.

The estimated costs of training peer mentors and delivering the intervention were £239 per mentor and £274 per participant respectively. The training costs would have been lower if fewer training events had been run with more mentors per event. This was not possible due to varying peer mentor availability and staggering of the mentor recruitment. A similar issue was noted in a study of peer mentorship for people with advanced cancer[54] and would need accounting for in a future RCT.

This trial included exploratory analyses of the impact of the OA peer mentorship intervention on patient-reported outcomes. These analyses provide a preliminary insight into the potential effectiveness of the intervention. However, the results must be interpreted cautiously because the present trial was not powered to detect statistically significant differences. The exploratory analyses suggest the OA peer mentorship intervention may improve self-management knowledge/behaviours (table 4). However, no significant effects of the OA peer mentorship intervention on other outcomes, such as pain and arthritis-related self-efficacy, were observed in the present trial.

A Cochrane review identified low to moderate quality evidence indicating self-management education programmes for people with OA do not result in any clinically meaningful benefits.[55] Despite this, the authors concluded that trials investigating alternative self-management support approaches, particularly those involving tailored support, are warranted. One such approach is the Staying Connected Programme, a tailored one-to-one 8-week self-management programme delivered by trained volunteers.[36] A recent quasi-experimental study identified significant improvements in pain and arthritis-related self-efficacy among individuals with arthritis who participated in this programme.[36] The Staying Connected Programme volunteers were not required to have an arthritis diagnosis, despite peer support being recognised as a valuable approach for supporting self-management.[11 12] In addition, the Staying Connected Programme aimed to support individuals with various types of arthritis and other long-term conditions.[36] The present trial's intervention therefore aimed to replicate some key elements of the Staying Connected Programme while also incorporating peer support and being tailored specifically to the needs of people with OA. Although there are disparities between the findings of the present trial and the Staying Connected Programme study, neither was an adequately powered RCT. Future work is therefore required to determine the effectiveness of both the present trial's OA peer mentorship intervention and the Staying Connected Programme.

In addition to drawing on the Staying Connected Programme, the development of the OA peer mentorship intervention incorporated multiple other sources and an expert review with key stakeholders. This approach, combined with extensive PPI, helped ensure

the intervention is feasible, acceptable and focused on the needs of individuals with OA. The substantial investment in the development process will also maximise the chances of the intervention proving effective during a future definitive RCT. A potential limitation is that the development process was not based on a single behaviour change theory or theoretical framework. However, the broad range of sources considered and input from multidisciplinary experts helped ensure that the intervention has a sound theoretical basis (figure 1). In particular, the focus on enhancing self-efficacy is consistent with other peer support interventions aimed at improving chronic condition self-management.[24 32 56 57]

This trial also presents additional limitations. Notably, 88% of participants were recruited through a physiotherapy service and hence had already received some self-management support. This could be addressed through using alternative recruitment sites, such as GP practices. Furthermore, the majority of participants were older, white, well-educated females, and a Cochrane review identified that the impact of self-management education programmes for OA may vary between Caucasian, educated females and other subgroups.[55] Additionally, peer support interventions may be most effective among the 'hardly reached', such as individuals with lower education levels.[58] Targeting the OA peer mentorship intervention to specific subgroups could therefore be valuable. Participants and peer mentors' experiences of the OA peer mentorship intervention were explored through a nested qualitative study (reported elsewhere). However, the impact of providing the intervention on the peer mentors was not quantitatively assessed. This is an important consideration because providing peer mentorship may have positive and/or negative effects on peer mentors.[11 26] Another limitation of the present trial was that most participants were aware of their group allocation when completing the baseline questionnaire, which may have influenced their questionnaire responses.[59] This could be overcome in a future definitive RCT by ensuring a greater number of trained peer mentors are available and/or administering the baseline questionnaire by post/online.

In conclusion, this trial's findings suggest peer mentorship is a feasible, acceptable and promising approach for improving OA self-management. Further investigation of the OA peer mentorship intervention is therefore warranted. However, minor modifications to the intervention and trial procedures should be considered, particularly regarding the participant and peer mentor recruitment procedures.

**Acknowledgements** The research team acknowledges the support of the National Institute for Health Research Clinical Research Network (NIHR CRN) and are grateful to all participants and peer mentors who were part of the study; and to the trial patient and public involvement (PPI) members and expert panel group.

**Contributors** GAM contributed to the conception and design of the study; acquisition, analysis and interpretation of data; and drafting the manuscript. LM, PGC, SRK and DA contributed to the design of the study. AMA contributed to the acquisition, analysis and interpretation of data and drafting the manuscript. ECL contributed to the acquisition and interpretation of data. ED-R contributed to the acquisition of data. GR contributed to the design of the study and the interpretation of data. TFM contributed to the analysis of data. All authors have been involved in revising the work for important intellectual content and have approved the final version for publication.

**Funding** This paper presents independent research funded by the National Institute for Health Research (NIHR) under its Research for Patient Benefit Programme (Grant Reference Number PB-PG-1215-20012). PGC and SRK are supported in part by the NIHR Leeds Biomedical Research Centre.

**Disclaimer** The views expressed are those of the authors and not necessarily those of the NIHR or the Department of Health and Social Care.

**Competing interests** None declared.

**Patient consent for publication** Not required.

**Ethics approval** Ethical approval was gained from the Greater Manchester South Research Ethics Committee (Reference:17/NW/0238). All participants provided written informed consent prior to participating in the trial.

**Provenance and peer review** Not commissioned; externally peer reviewed.

**Data availability statement** Data are available on reasonable request. Deidentified participant data and the trial protocol are available from the corresponding author via A.Anderson@leeds.ac.uk on reasonable request. Reuse is permitted for the purpose of health and care research as long as the original creators are acknowledged.

**ORCID iDs**
Anna M Anderson http://orcid.org/0000-0002-4048-6880
Teumzghi F Mebrahtu http://orcid.org/0000-0003-4821-2304
Philip G Conaghan http://orcid.org/0000-0002-3478-5665

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
