## [Reviewer comments · BMJ Open]

ARTICLE DETAILS

TITLE (PROVISIONAL)	Peer mentorship to improve self-management of hip and knee osteoarthritis: A randomised feasibility trial
AUTHORS	Anderson, Anna; Lavender, Elizabeth; Dusabe-Richards, Esther; Mebrahtu, Teumzghi; McGowan, Linda; Conaghan, Philip; Kingsbury, Sarah; Richardson, Gerry; Antcliff, Deborah; McHugh, Gretl

VERSION 1 – REVIEW

REVIEWER	Suzanne McDonough University of Ulster, Health and Rehab Sciences Research Institute
REVIEW RETURNED	25-Nov-2020

GENERAL COMMENTS	This is well written report of a feasibility study that has been conducted with rigor. I have made some suggestions below for the authors to consider. The description of the intervention development needs to be enhanced particularly given the study aim to 'develop the intervention', so needs more detail in the methods and the results. I query the use of change data in PHI for your sample size, and would like less emphasis on discussion of clinical end points given the very small size of the study. Line 183-is there a reference for the rapid review? This is important as it would appear that the main source of evidence for the intervention comes from one small study (ref 23) 191-194-what was the process for refinement, and what were main changes made? It would be important to understand how the intervention was informed by the literature, and its theoretical underpinning. I would recommend https://www.ncbi.nlm.nih.gov/books/NBK540944/ on development of a peer led intervention for older adults. https://implementationscience.biomedcentral.com/articles/10.1186/s13012-016-0418-2 is also a very detailed paper on the development of a self management intervention in people with arthritis. 253-recorded in writing or how? Line 269-provide a rationale for using PIH change scores from your feasibility for the sample size in a main trial, as my understanding is that this is not recommended (https://pilotfeasibilitystudies.biomedcentral.com/articles/10.1186/s40814-019-0493-7) Lines 277-not clear why outcomes were measured after person being informed of their allocation? I would prefer not to see inferential stats on the outcome data given the very small sample size e.g. line 502, and value of discussion section on this preliminary analysis is very limited, and the comparison with the Staying Connected study. Line 541, presumably you may include other regions in a multi centre trial so could recruit from a wider geographical spread for
--

	peer mentors?
--	---------------

REVIEWER	Eva Ekvall Hansson Lund University, Department of Health Sciences
REVIEW RETURNED	12-Feb-2021

GENERAL COMMENTS	General comments This manuscript is well written and the topic is relevant, since OA is increasing in the population. I have a few comments regarding method and how the results are displayed. Please find my specific comments below. Title Relevatn. Abstract Relevant Strengths and limitations First point: this is a description of what was done, not a strength or a limitation Introduction Appropriate. Aim Well described Method Page 5, line 113: please add a reference to where the nested qualitative study has been reported. Page 6, line 126: please write the full name the first time an abbreviation is used (NHS). Page 9, line 204 and line 209: One compulsory two-day training event is mentioned as well as three training events. Are these the same or different events? What comprise the events of? Page 9, line 205: Please explain what "Disclosure and Barring Service checks is, with reference. Page 10, line 225-228: please include examples of content in the sessions. Page 13, line 270-273: is this a fotnot to table 1? Page 15, line 324: The five dimensions in EQ5D is not scored on a 1-5 Likert scale. EQ5D comprise of questions that each has five possible answers, the answers is then converted do numbers which in turn is converted to the index. Please also include the reference by van Hunt that EuroQol recommend. Results Effect size is used in the results. A description of how effect size was calculated should be included in the method section. Discussion Page 28, line 567: "no significant effects on pain and ASES". Patient Education programmes has shown to have effect on both pain and ASES, perhaps rephrase to clarify that it was the addition of peer-mentorship to the programme that did not have any effect on pain and ASES. Page 29, line 599: include reference to where the nested quality study is reported.
--

	Conclusion Adequate Tables and figures Table 2: remove brackets from "n" Table 4: How is "effect" measured? Do you mean effect size? Add information in the footnote on how effect or effect size is measured. References Add reference about the nested quality study
--	---

REVIEWER	K Cooper Robert Gordon University, School of Health Sciences
REVIEW RETURNED	20-Feb-2021

GENERAL COMMENTS	Well done on a very well written manuscript reporting your interesting and important feasibility trial of peer mentorship to improve self-management of hip & knee OA. I just have a couple of comments that may enhance the manuscript:  1. Recruiting and screening potential peer mentors is a challenging aspect of this type of work. You report that those "assessed as suitable" were invited to the training (line 204) and also report that 11 were excluded due to "not meeting eligibility criteria" (Suppl figure 1). Can you provide any further details on this for the reader? Were the criteria only being aged 50+ and having hip/knee OA, or were any other strategies used during screening or training the peer mentors to determine their eligibility? Some previous studies have required participants to take a knowledge test or used observation of interpersonal skills as additional criteria. Further detail on how you identified people to be suitable as peer mentors would be helpful for others doing work in this field. 2. Although the intervention is described well, I was surprised by the lack of underpinning theory, particularly when the overall aim is to change health behaviour. Presumably the rapid review of published literature identified several theory-based approaches to peer mentoring interventions. Perhaps something could be added to provide the reader with an understanding of the proposed mechanisms of action of the intervention.
--

VERSION 1 – AUTHOR RESPONSE

Reviewer: 1
Prof. Suzanne McDonough, University of Ulster

Comments to the Author:

This is well written report of a feasibility study that has been conducted with rigor. I have made some suggestions below for the authors to consider. The description of the intervention development needs to be enhanced particularly given the study aim to 'develop the intervention', so needs more detail in the methods and the results. I query the use of change data in PHI for your sample size, and would like less emphasis on discussion of clinical end points given the very small size of the study.

Line 183-is there a reference for the rapid review? This is important as it would appear that the main source of evidence for the intervention comes from one small study (ref 23)

We did not publish the rapid review but we have now added further details about the rapid review and the intervention development process.

'The development and feasibility testing of the OA peer mentorship intervention was guided by the Medical Research Council guidance on developing and evaluating complex interventions [18].'
(Methods; Page 5; Lines 119-121)

'The OA peer mentorship intervention was developed in two stages. The first stage included a rapid review of published primary studies investigating one-to-one peer support interventions. The review aimed to identify: the range of methods and approaches used in delivering peer support interventions; training and support approaches for peer mentors involved in intervention delivery; and challenges of developing and implementing face-to-face peer support to improve self-management of long-term conditions. Using Medline, CINAHL & PsycInfo, a search of the literature between 2007 and 2018 was undertaken using the terms and synonyms 'peer support', 'long term condition' and 'intervention'. Thirteen papers were included [16, 24-35]. The findings highlighted the importance of encouraging a person-centred approach and retaining flexibility within the peer mentorship sessions. There was little information about 'matching' the mentee with the peer mentor, although some studies based this on factors such as gender and age. There were issues around recruitment of peer mentors and a need to provide support and guidance to peer mentors throughout the intervention.

A preliminary version of the OA peer mentorship intervention was developed based on the rapid review findings and the following sources:

- Guidelines on self-management and OA from organisations such as the United Kingdom (UK) National Institute for Health and Care Excellence (NICE) and the UK charity Versus Arthritis.
- The 'Staying Connected Programme' – an arthritis self-management programme previously run by Arthritis Care Northern Ireland (now Versus Arthritis Northern Ireland).[36]
- Input from project team members, including a consultant rheumatologist, a health psychologist and a physiotherapist specialising in activity pacing/chronic pain management.

The above sources were also used to develop a draft peer mentor educational resource pack. The pack was designed to supplement peer mentor training sessions and be used as a resource during mentorship sessions. The pack included a range of handouts that peer mentors could give to participants.' (Methods; Pages 8-9; Lines 181-121)

191-194-what was the process for refinement, and what were main changes made? It would be important to understand how the intervention was informed by the literature, and its theoretical underpinning. I would recommend <https://www.ncbi.nlm.nih.gov/books/NBK540944/> on development of a peer led intervention for older adults.

<https://implementationscience.biomedcentral.com/articles/10.1186/s13012-016-0418-2> is also a very detailed paper on the development of a self management intervention in people with arthritis.

We have now added further details about how the intervention was refined.

'The second stage of the development process consisted of expert review sessions conducted with the following key stakeholders: study Patient and Public Involvement (PPI) members (n=2); other older individuals with OA (n=5); health professionals (n=4); voluntary/community organisation representatives (n=4); and researchers (n=2). Review sessions were conducted face-to-face (group, paired or individual meetings), via telephone or via email. The stakeholders were provided with information about the provisional OA peer mentorship intervention and a copy of the draft educational pack. Stakeholders' opinions of these were assessed using a pro forma.

Key refinements made based on the expert review sessions included:

- Peer mentors were encouraged not cover too many topics in the first mentorship session to allow more time for developing rapport and managing participant expectations.

- Participants were provided with notebooks for recording goals, reflections and additional notes.
- The language used in the educational pack was simplified.
- Additional infographic handouts and further information on falls, local authority services and emotional well-being were added to the educational resource pack.' (Methods; Pages 9-10; Lines 214-233)

We have also provided further details about the content of the intervention and provided a logic model of the intervention.

'The finalised OA peer mentorship intervention aimed to improve participants' health outcomes through increasing their engagement with self-management behaviours. Figure 1 presents a logic model of the intervention, including the proposed mechanisms of action. The intervention involved up to eight one-hour self-management support sessions delivered by a trained peer mentor. During the sessions, the peer mentor provided guided support that incorporated multiple behaviour change techniques (BCTs) and covered a combination of core and optional topics (figure 1). In line with a person-centred approach, the implementation of BCTs and the choice and order of topics covered was flexible and participant-led. However, peer mentors were encouraged to cover all core topics at least once and set/review goals with the participant weekly. Online supplementary table 3 provides examples of the implementation of the key BCTs employed.' (Methods; Pages 10-11; Lines 237-248)

Furthermore, we have addressed the intervention development process in the discussion.

'In addition to drawing on the Staying Connected Programme, the development of the OA peer mentorship intervention incorporated multiple other sources and an expert review with key stakeholders. This approach, combined with extensive PPI, helped ensure the intervention is feasible, acceptable and focused on the needs of individuals with OA. The substantial investment in the development process will also maximise the chances of the intervention proving effective during a future definitive RCT. A potential limitation is that the development process was not based on a single behaviour change theory or theoretical framework. However, the broad range of sources considered and input from multidisciplinary experts helped ensure that the intervention has a sound theoretical basis (figure 1). In particular, the focus on enhancing self-efficacy is consistent with other peer support interventions aimed at improving chronic condition self-management.[24, 32, 56, 57]' (Discussion; Pages 31-32; Lines 647-658)

253-recorded in writing or how?

We have now added details of the recording format.

'Peer mentors completed a 'session summary' in writing following each session, detailing the topics covered during that session, any challenges encountered and their reflections on the progress made.' (Methods; Page 11; Lines 260-263)

Line 269-provide a rationale for using PIH change scores from your feasibility for the sample size in a main trial, as my understanding is that this is not recommended (<https://pilotfeasibilitystudies.biomedcentral.com/articles/10.1186/s40814-019-0493-7>)

Thank you for highlighting this interesting paper. We agree with the authors' conclusions that the decision about whether or not to proceed with a definitive trial should not be based solely on the effect sizes and confidence intervals of the feasibility trial. In line with this, we pre-specified four success criteria for proceeding to a definitive trial (Methods; Page 14; Table 1) based on relevant guidance (<https://bmcmedresmethodol.biomedcentral.com/articles/10.1186/1471-2288-10-1>). These correspond

with the key objectives of our feasibility trial, one of which was to generate the sample size needed for a definitive RCT.

Given the OA peer mentorship intervention we investigated has not been studied previously, we used the PIH scores from our feasibility trial for the sample size calculation. We have now clarified why we chose the PIH scores specifically.

'The PIH scale scores were used because the PIH scale was chosen as the provisional primary outcome measure.' (Methods; Page 14; Table 1).

Lines 277-not clear why outcomes were measured after person being informed of their allocation?

We have now added further details about this.

'This approach was chosen due to the fluctuating availability of trained peer mentors. This meant there was sometimes a delay between participants being allocated to the intervention group and being able to commence their mentorship sessions. Therefore, most intervention group participants completed the baseline questionnaire with the Volunteer Coordinator immediately prior to their initial mentorship session to avoid a delay between completion of the baseline questionnaire and commencement of their mentorship sessions.' (Methods; Page 15; Lines 323-329)

We have also acknowledged this as a limitation in the discussion and highlighted how it could be addressed in a future trial.

'Another limitation of the present trial was that most participants were aware of their group allocation when completing the baseline questionnaire, which may have influenced their questionnaire responses.[59] This could be overcome in a future definitive RCT by ensuring a greater number of trained peer mentors are available and/or administering the baseline questionnaire by post/online.' (Discussion; Pages 32-33; Lines 674-677)

I would prefer not to see inferential stats on the outcome data given the very small sample size e.g. line 502, and value of discussion section on this preliminary analysis is very limited, and the comparison with the Staying Connected study.

Thank you for this comment. We agree that caution is required when using inferential statistics on the outcome data. However, we believe that including the results of the exploratory analyses is worthwhile to provide readers with a preliminary insight into the potential effectiveness of the intervention. We have now expanded the discussion section on the preliminary analysis to clarify this.

'This trial included exploratory analyses of the impact of the OA peer mentorship intervention on patient-reported outcomes. These analyses provide a preliminary insight into the potential effectiveness of the intervention. However, the results must be interpreted cautiously because the present trial was not powered to detect statistically significant differences.' (Discussion; Page 30; Lines 616-620).

We have also amended the discussion of the Staying Connected Programme.

'One such approach is the Staying Connected Programme – a tailored one-to-one eight-week self-management programme delivered by trained volunteers.[36] A recent quasi-experimental study identified significant improvements in pain and arthritis-related self-efficacy amongst individuals with arthritis who participated in this programme.[36] The Staying Connected Programme volunteers were not required to have an arthritis diagnosis, despite peer support being recognised as a valuable

approach for supporting self-management.[11, 12] In addition, the Staying Connected Programme aimed to support individuals with various types of arthritis and other long-term conditions [36]. The present trial's intervention therefore aimed to replicate some key elements of the Staying Connected Programme whilst also incorporating peer support and being tailored specifically to the needs of people with OA. Although there are disparities between the findings of the present trial and the Staying Connected Programme study, neither was an adequately powered RCT. Future work is therefore required to determine the effectiveness of both the present trial's OA peer mentorship intervention and the Staying Connected Programme.' (Discussion; Page 31; Lines 630-645)

Line 541, presumably you may include other regions in a multi centre trial so could recruit from a wider geographical spread for peer mentors?

We agree that recruiting peer mentors from a wider geographical area would be helpful for a future trial and have now clarified that in the discussion.

'These could include using multiple recruitment sites across a wider geographical area, recruiting mentors over a longer time-period, using snowballing recruitment with previously trained peer mentors and optimising recruitment through media outlets, such as magazine advertisements.' (Discussion; Page 29; Lines 592-595)

Reviewer: 2

Dr. Eva Ekvall Hansson, Lund University

Comments to the Author:

General comments

This manuscript is well written and the topic is relevant, since OA is increasing in the population. I have a few comments regarding method and how the results are displayed.

Please find my specific comments below.

Title

Relevant.

Abstract

Relevant

Strengths and limitations

First point: this is a description of what was done, not a strength or a limitation

We believe that developing a novel intervention is an important strength of this study. We have now amended the wording to help clarify this.

'This randomised feasibility trial is the first study to develop a novel peer mentorship intervention specifically focused on improving self-management of hip and knee osteoarthritis (OA).' (Strengths and Limitations; Page 3; Lines 58-60)

Introduction

Appropriate.

Aim

Well described

Method

Page 5, line 113: please add a reference to where the nested qualitative study has been reported.

Our manuscript on the qualitative evaluation of the peer mentorship intervention has been accepted by Disability and Rehabilitation on the condition that this feasibility trial manuscript is accepted for publication. We are also currently preparing a manuscript on the qualitative interviews conducted with intervention group participants. We cannot therefore reference the nested qualitative study yet. We will however ensure that the present feasibility paper is referenced in the two planned publications of the nested qualitative study.

Page 6, line 126: please write the full name the first time an abbreviation is used (NHS).

We have now added this.

'Potential participants were identified from rheumatology and orthopaedic clinics of one secondary care National Health Service (NHS) Trust and physiotherapy clinics and electronic records of one primary care NHS Trust.' (Methods; Page 6; Lines 127-129).

Page 9, line 204 and line 209: One compulsory two-day training event is mentioned as well as three training events. Are these the same or different events? What comprise the events of?

We have now clarified why we held three training events.

'Three training events were held due to varying peer mentor availability and staggering of the mentor recruitment, which meant it was not possible to train all mentors in one event.' (Methods; Page 12; Lines 281-283)

We have included a brief summary of the training events in the main manuscript text. Details of the specific areas covered are available in supplementary table 4.

'The training events involved presentations and interactive activities covering OA self-management topics, mentorship skills and the practicalities of the peer mentor role (online supplementary table 4).' (Methods; Page 12; Lines 291-293)

In addition, we have written a separate manuscript reporting a qualitative evaluation of the peer mentorship intervention. The manuscript provides further details about the peer mentor training events and has been accepted by Disability and Rehabilitation on the condition that this feasibility trial manuscript is accepted for publication.

We would also like to highlight that we have made a minor change to the peer mentor flow diagram in supplementary figure 1 by adding a footnote to explain the following.

'One trained peer mentor who attended Training event 1 also attended Training event 3 as refresher training.' (Supplementary figure 1)

Page 9, line 205: Please explain what "Disclosure and Barring Service checks is, with reference.

We have now provided an explanation and reference for these checks.

'Prior to being matched with a participant, all peer mentors were required to complete enhanced Disclosure and Barring Service (criminal record) checks,[24] for safeguarding purposes.' (Methods; Page 12; Lines 275-277)

Page 10, line 225-228: please include examples of content in the sessions.

We have now provided further details of the content of the sessions in the text and logic model.

'During the sessions, the peer mentor provided guided support that incorporated multiple behaviour change techniques (BCTs) and covered a combination of core and optional topics (figure 1). In line with a person-centred approach, the implementation of BCTs and the choice and order of topics covered was flexible and participant-led. However, peer mentors were encouraged to cover all core topics at least once and set/review goals with the participant weekly. Online supplementary table 3 provides examples of the implementation of the key BCTs employed.' (Methods; Pages 10-11; Lines 241-248)

Page 13, line 270-273: is this a footnote to table 1?

Yes – we have now labelled this as 'Table 1 Footnote' for clarity (Methods; Page 15; Line 311). We have also labelled the other table footnotes for consistency.

Page 15, line 324: The five dimensions in EQ5D is not scored on a 1-5 Likert scale. EQ5D comprise of questions that each has five possible answers, the answers is then converted do numbers which in turn is converted to the index.

Thank you for highlighting this. We have updated the description of the EQ-5D-5L accordingly.

'The descriptive system includes five dimensions, each of which has five response levels. Each response is converted to a single-digit number. The numbers for each of the five dimensions can then be converted to a single index value anchored at 0 (a state equivalent to dead) and 1 (full health).' (Methods; Page 17; Lines 369-375)

Please also include the reference by van Hunt that EuroQol recommend.

There is no van Hunt reference in the EQ-5D-5L user guide or on the EuroQol 'EQ-5D-5L Key references' webpage (<https://euroqol.org/publications/key-euroqol-references/eq-5d-5l/>). We have added a reference by van Hout in case that is what you are referring to, as well as leaving in our original reference that is listed on the EuroQol 'EQ-5D-5L Key references' webpage.

48. Herdman M, Gudex C, Lloyd A, et al. Development and preliminary testing of the new five-level version of EQ-5D (EQ-5D-5L). *Qual Life Res* 2011;20(10):1727-36. doi: 10.1007/s11136-011-9903-x.
49. van Hout B, Janssen MF, Feng YS et al. Interim scoring for the EQ-5D-5L: mapping the EQ-5D-5L to EQ-5D-3L value sets. *Value Health* 2012;15(5):708-15. doi: 10.1016/j.jval.2012.02.008.' (Reference list).

Results

Effect size is used in the results. A description of how effect size was calculated should be included in the method section.

We have described how the effect sizes were calculated in the Methods section.

'Between-group comparisons were made using ANCOVA models and, where the scores were significantly skewed, quantile regression models were used.[38]' (Methods; Pages 18; Lines 387-389)

Discussion

Page 28, line 567: “no significant effects on pain and ASES”. Patient Education programmes has shown to have effect on both pain and ASES, perhaps rephrase to clarify that it was the addition of peer-mentorship to the programme that did not have any effect on pain and ASES.

We have now clarified this.

‘However, no significant effects of the OA peer mentorship intervention on other outcomes, such as pain and arthritis-related self-efficacy, were observed in the present trial.’ (Discussion; Page 30; Lines 622-624)

Page 29, line 599: include reference to where the nested quality study is reported.

As explained above, the nested qualitative study is not yet published so we are not able to reference it at present. We will however ensure that the present feasibility paper is referenced in the two planned publications of the nested qualitative study.

Conclusion
Adequate

Tables and figures

Table 2: remove brackets from “n”

We have now removed the brackets.

Table 4: How is “effect” measured? Do you mean effect size? Add information in the footnote on how effect or effect size is measured.

We have now clarified that we mean ‘effect size’ (Results; Page 26; Table 4). Descriptions of how the outcome measures are scored are provided in the Methods section.

‘Provisional primary outcome measure

- Revised 12-Item Partners in Health (PIH) Scale: 12-item scale that assesses chronic condition self-management.[41] Each item is scored on a 9-point Likert scale (0-8). Scores are summed to give a total score (0-96). Higher scores indicate higher self-management knowledge/behaviours.

Provisional secondary outcome measures

- Multidimensional Scale of Perceived Social Support (MSPSS): 12-item scale that assesses perceived social support from a significant other, family and friends.[42] Each item is scored on a 7-point Likert scale (1-7) and the mean for all items is calculated. Higher scores indicate greater perceived social support.
- Western Ontario and McMaster’s University Osteoarthritis Index Likert version (WOMAC® 3.1): 24-item disease-specific scale with three subscales that assess pain (5 items), stiffness (2 items) and physical function (17 items).[43] Each item is scored on a 5-point Likert scale (0-4). Scores are summed to give a total score (0-96) and subscale scores (pain: 1-20; stiffness: 0-8; physical function: 0-68). Higher scores indicate more severe problems.
- 8-item Arthritis Self-Efficacy Scale English Version (ASES-8): 8-item scale that assesses self-efficacy for managing arthritis symptoms.[44-46] Each item is scored on a 10-point Likert scale (1-10) and the mean for all items is calculated. Higher scores indicate greater self-efficacy.
- Hospital Anxiety and Depression Scale (HADS): 14-item scale with two subscales that assess symptoms of anxiety (7 items) and depression (7 items).[47] Each item is scored on a 4-point Likert

scale (0-3). Scores are summed to give a total score (0-42) and subscale scores (0-21). Higher scores indicate more severe symptoms.

• EQ-5D-5L: Descriptive system and visual analogue scale (VAS) that assess general health status.[48, 49] The descriptive system includes five dimensions, each of which has five response levels. Each response is converted to a single-digit number. The numbers for each of the five dimensions can then be converted to a single index value anchored at 0 (a state equivalent to dead) and 1 (full health). The VAS consists of a single score on a scale from 0 (worst health imaginable) to 100 (best health imaginable).’ (Methods; Pages 16-17; Lines 339-375)

References

Add reference about the nested quality study

As explained above, the nested qualitative study is not yet published so we are not able to reference it at present. We will however ensure that the present feasibility paper is referenced in reports of the nested qualitative study.

Reviewer: 3

Dr. K Cooper, Robert Gordon University

Comments to the Author:

Well done on a very well written manuscript reporting your interesting and important feasibility trial of peer mentorship to improve self-management of hip & knee OA.

I just have a couple of comments that may enhance the manuscript:

1. Recruiting and screening potential peer mentors is a challenging aspect of this type of work. You report that those "assessed as suitable" were invited to the training (line 204) and also report that 11 were excluded due to "not meeting eligibility criteria" (Suppl figure 1). Can you provide any further details on this for the reader? Were the criteria only being aged 50+ and having hip/knee OA, or were any other strategies used during screening or training the peer mentors to determine their eligibility? Some previous studies have required participants to take a knowledge test or used observation of interpersonal skills as additional criteria. Further detail on how you identified people to be suitable as peer mentors would be helpful for others doing work in this field.

Thank you for highlighting this important consideration. We have covered this only briefly in the present feasibility trial manuscript because we have written a separate manuscript reporting a qualitative evaluation of the peer mentorship intervention. The qualitative manuscript provides further details about the peer mentor recruitment process and has been accepted by Disability and Rehabilitation on the condition that this feasibility trial manuscript is accepted for publication.

Peer mentors were required to meet basic eligibility criteria (aged ≥ 50 years old, have hip and/or knee OA and be able to travel independently). We have now clarified this in the manuscript text.

‘Peer mentors were trained volunteers aged ≥ 50 years old with hip and/or knee OA who were able to travel independently.’ (Methods; Page 11; Lines 267-268)

In addition, suitability for the role (including appropriate interpersonal skills) was informally assessed both by the volunteers themselves and by the study team during the application process and training events. We did not require volunteers to undertake a knowledge test because the compulsory two-day training event focused on equipping volunteers with the knowledge and skills required for the peer mentor role.

2. Although the intervention is described well, I was surprised by the lack of underpinning theory, particularly when the overall aim is to change health behaviour. Presumably the rapid review of

published literature identified several theory-based approaches to peer mentoring interventions. perhaps something could be added to provide the reader with an understanding of the proposed mechanisms of action of the intervention.

Thank you for this suggestion. We have now added further details about the proposed mechanisms of action of the intervention in the manuscript text and added a logic model of the intervention.

'The finalised OA peer mentorship intervention aimed to improve participants' health outcomes through increasing their engagement with self-management behaviours. Figure 1 presents a logic model of the intervention, including the proposed mechanisms of action. The intervention involved up to eight one-hour self-management support sessions delivered by a trained peer mentor. During the sessions, the peer mentor provided guided support that incorporated multiple behaviour change techniques (BCTs) and covered a combination of core and optional topics (figure 1). In line with a person-centred approach, the implementation of BCTs and the choice and order of topics covered was flexible and participant-led. However, peer mentors were encouraged to cover all core topics at least once and set/review goals with the participant weekly. Online supplementary table 3 provides examples of the implementation of the key BCTs employed.' (Methods; Pages 10-11; Lines 237-248)

In addition, we have also now addressed the intervention development in the discussion.

'In addition to drawing on the Staying Connected Programme, the development of the OA peer mentorship intervention incorporated multiple other sources and an expert review with key stakeholders. This approach, combined with extensive PPI, helped ensure the intervention is feasible, acceptable and focused on the needs of individuals with OA. The substantial investment in the development process will also maximise the chances of the intervention proving effective during a future definitive RCT. A potential limitation is that the development process was not based on a single behaviour change theory or theoretical framework. However, the broad range of sources considered and input from multidisciplinary experts helped ensure that the intervention has a sound theoretical basis (figure 1). In particular, the focus on enhancing self-efficacy is consistent with other peer support interventions aimed at improving chronic condition self-management.[24, 32, 56, 57]' (Discussion; Pages 31-32; Lines 647-658)

VERSION 2 – REVIEW

REVIEWER	Suzanne McDonough University of Ulster, Health and Rehab Sciences Research Institute
REVIEW RETURNED	13-May-2021

GENERAL COMMENTS	I am happy the authors have addressed my comments.
--

REVIEWER	Eva Ekvall Hansson Lund University, Department of Health Sciences
REVIEW RETURNED	16-Mar-2021

GENERAL COMMENTS	General comments This manuscript is well written and the topic is relevant, since OA is increasing in the population. I have a few comments regarding method and how the results are displayed. Please find my specific comments below. Title Relevant. Abstract
---

	Relevant Strengths and limitations First point: this is a description of what was done, not a strength or a limitation Introduction Appropriate. Aim Well described Method Page 5, line 113: please add a reference to where the nested qualitative study has been reported. Page 6, line 126: please write the full name the first time an abbreviation is used (NHS). Page 9, line 204 and line 209: One compulsory two-day training event is mentioned as well as three training events. Are these the same or different events? What do the events comprise of? Page 9, line 205: Please explain what "Disclosure and Barring Service checks is, with reference. Page 10, line 225-228: please include examples of content in the sessions. Page 13, line 270-273: is this a footnote to table 1? Page 15, line 324: The five dimensions in EQ5D is not scored on a 1-5 Likert scale. EQ5D comprise of questions that each has five possible answers, the answers are then converted do numbers which in turn is converted to the index. Please also include the reference by van Hunt that EuroQol recommend. Results Effect size is used in the results. A description of how effect size was calculated should be included in the method section. Discussion Page 28, line 567: "no significant effects on pain and ASES". Patient Education programmes has shown to have effect on both pain and ASES, perhaps rephrase to clarify that it was the addition of peer-mentorship to the programme that did not have any effect on pain and ASES. Page 29, line 599: include reference to where the nested quality study is reported. Conclusion Adequate Tables and figures Table 2: remove brackets from "n" Table 4: How is "effect" measured? Do you mean effect size? Add information in the footnote on how effect or effect size is measured. References Add reference about the nested quality study
--	---

REVIEWER	K Cooper Robert Gordon University, School of Health Sciences
REVIEW RETURNED	29-Mar-2021

GENERAL COMMENTS	Thank you for addressing my previous comments, the description of the theoretical underpinning for the intervention and the logic model in particular are very positive additions to this manuscript.
---